# The Study of Magnetoimpedance Effect for Magnetoelectric Laminate Composites with Different Magnetostrictive Layers

**DOI:** 10.3390/ma14216397

**Published:** 2021-10-25

**Authors:** Lei Chen, Yao Wang, Tianhong Luo, Yongkang Zou, Zhongjie Wan

**Affiliations:** 1Key Lab of Computer Vision and Intelligent Information System, Chongqing University of Arts and Sciences, Chongqing 402160, China; 67638515@163.com (L.C.); future_wan@foxmail.com (Z.W.); 2School of Electronic Information and Electrical Engineering, Shanghai Jiao Tong University, Shanghai 200240, China

**Keywords:** magnetostrictive material, magnetoimpedance effect, magnetostrictive strain, magnetic permeability, Young’s modulus, magnetoelectric composite

## Abstract

The rectangular magnetoelectric (ME) composites of Metglas/PZT and Terfenol-D/PZT are prepared, and the effects of a magnetostrictive layer’s material characteristics on the magnetoimpedance of ME composite are discussed and experimentally investigated. The theoretical analyses show that the impedance is not only dependent on Young’s modulus and the magnetostrictive strain of magnetostrictive material but is also influenced by its relative permeability. Compared with Terfenol-D, Metglas possesses significantly higher magnetic permeability and larger magnetostrictive strain at quite low H_dc_ due to the small saturation field, resulting in the larger magnetoimpedance ratio. The experimental results demonstrate that the maximum magnetoimpedance ratios (i.e., ΔZ/Z) of Metglas/PZT composite are about 605.24% and 239.98% at the antiresonance and resonance, respectively. Specifically, the maximum ΔZ/Z of Metglas/PZT is 8.6 times as high as that of Terfenol-D/PZT at the antiresonance. Such results provide the fundamental guidance in the design and fabrication of novel multifunction devices based on the magnetoimpedance effect of ME composites.

## 1. Introduction

The magnetoelectric (ME) laminate composites consisting of magnetostrictive and piezoelectric materials have gained intense research interests due to their applications in multifunctional devices such as memory devices, tunable inductors, magnetic sensors, electrostatically tunable filters and spin-charge transducers, etc. [1,2,3,4,5,6,7,8,9,10]. ME composites are especially promising for tunable electrical component (resistors, capacitors, inductors, etc.) applications [11,12,13,14], among which E-field tunable inductors based on ME laminate composites have been widely studied recently. Fang et al. have reported the electric-field-induced inductance change for a heterogeneous composite consisting of a PZT bar embedded in a MnZn ferrite ring [15]. Lou et al. experimentally studied the electrostatically tunable magnetoelectric inductors with multiferroic composite cores consisting of Metglas/lead zirconate titanate/Metglas [16]. Zhang et al. proposed a tunability-improved ME inductor in the symmetrical composite consisting of Ni_0.8_Zn_0.2_Fe_2_O_4_ platelet and piezoelectric ceramics Pb(Zr,Ti)O_3_ slab with laminate Metglas foils [17]. Additionally, DC magnetic field (H_dc_) tuning of electrical components has attracted much attention and become an exciting research topic. Singh et al. investigated the giant magnetocapacitance of magnetoelectric Bi_0.5_Na_0.5_TiO_3_/NiFe_2_O_4_ composite at a high magnetic field [18]. Zhang et al. studied the effect of boundary conditions on the magnetocapacitance of a ring-type magnetoelectric structure [19]. Wang et al. have also reported the large room-temperature magnetocapacitance in the Tb_x_Dy_1−x_Fe_2−y_/PbZr_x_Ti_1−x_O_3_/Tb_x_Dy_1−x_Fe_2−y_ laminate at a saturated magnetic field of 1.5 kOe [13].

The magnetically tunable capacitance, inductance and impedance effects provide a promising application for sensors and transducers, etc. However, few articles have discussed tuning the giant magnetoimpedance (MI) effect of ME composites with the DC magnetic field at room temperature [20,21]. Additionally, compared to the conventional magnetoelectric (ME) effect, the MI effect of ME composite can be obtained by applying only the DC magnetic field (H_dc_) without superimposing the alternating magnetic field (H_ac_), which decreases the power consumption and facilitates the miniaturization of the ME device. Furthermore, the analysis and comparisons of various magnetostrictive material’s influences on the MI effect of ME composites are rarely reported, which hinders the design and optimization of tunable magnetoimpedance devices. Correspondingly, to facilitate the practical device applications, it is both physically interesting and technologically important to study and understand the magnetoimpedance effect of ME composites with different magnetostrictive materials.

In this work, the magnetoimpedance (MI) effects of ME laminate composites with different magnetostrictive materials are analyzed and experimentally investigated. It is interesting to find that the Metglas/PZT laminate composite demonstrates the significantly higher MI effect compared to Terfenol-D/PZT laminate composite, whose maximum impedance ratio is 8.6 times as high as that of Terfenol-D/PZT at the antiresonance frequency. Furthermore, the influences of different magnetostrictive materials (e.g., Metglas and Terfenol-D) on the MI effect of ME composites are analyzed and corresponding resonant frequencies are explored. It is interesting to find that the effective magnetic permeability, magnetostrictive strain and Young’s modulus of magnetostrictive materials play critical roles in improving the magnetoimpedance effect of ME composites.

## 2. Experiment

The rectangular Metglas/PZT and Terfenol-D/PZT bilayer composites were prepared to investigate the magnetoimpedance effect of ME composite. The PZT plate with dimensions of 12 × 6 × 0.8 mm^3^ was supplied by Electronics Technology Group Corporation No.26 Research Institute, Chongqing, China, which exhibits the high ferroelectric Curie temperature and piezoelectric constant. The Ag electrodes of PZT were distributed on the top and bottom surfaces, and the PZT was polarized with an electric field of 25 kV/cm along the thickness direction. Terfenol-D with giant saturation magnetostriction (~1200 ppm) was purchased from Gansu Tianxing Rare Earth Functional Materials Co., Ltd., Gansu, China. Then the Terfenol-D layer was cut into a rectangular plate with a length of 12 mm, width of 6 mm and thickness of 1 mm. The soft magnetostrictive Metglas 2605SA1 (i.e., FeBSiC) with extremely high relative permeability (μ_r_ = 50,000) and good mechanical properties was produced by Foshan Huaxin Microlite Metal Co., Ltd., Foshan, China, and the size is 12 × 6 × 0.03 mm^3^. To fabricate the ME samples, the Terfenol-D plates were dipped into acetone to clean the surface oxidation layer at first. Then, the piezoelectric layer and magnetostrictive layer were bonded together with epoxy adhesive, which were pressed using a hydraulic press and cured at 80 °C for 4 h to minimize the epoxy thickness between layers.

The external DC magnetic field (H_dc_) was generated by a pair of annular permanent magnets (Nd–Fe–B), which is along the longitudinal direction of the ME composite. The H_dc_ between 0 and 1500 Oe was calibrated by a Gaussmeter. Then the impedance of ME composites was measured with a precision impedance analyzer (4194 A HP Agilent, Santa Clara, CA, USA), as shown in Figure 1. Specifically, the impedance spectrum was measured at different DC magnetic fields by using a frequency-swept method around the resonance frequency. It is noted that before the impedance measurements, the standard calibration was performed under open and short circuit conditions to eliminate the inherent features of the measured system associated with the connecting cable and background circuit. Furthermore, magnetization hysteresis (M–H) and strain coefficients were measured with the vibrating sample magnetometer (VSM) and laser Doppler vibrometer (Polytec OFV-5000, Berlin, Germany), respectively.

## 3. Results and Discussion

To investigate the magnetoimpedance of ME composites, the impedance of Metglas/PZT composite was measured as a function of electrical excitation frequency under different longitudinal DC magnetic fields, as shown in Figure 2. It is interesting to find that both the antiresonance frequency f_a_ corresponding to the maximum impedance (Z_m_) and the resonance frequency f_r_ corresponding to the minimum impedance (Z_n_) show strong dependences on H_dc_, shown as the insets of Figure 3. f_r_ and f_a_ vary with the DC magnetic field in a similar trend for the ME composite. For Metglas/PZT composite, f_r_ and f_a_ increase quickly with the increased DC magnetic field until the magnetization of Metglas reaches saturation. The resonance frequency f_r_ varies from 138.3 to 139.3 kHz when H_dc_ increases from 0 Oe to 600 Oe. Such changes of f_r_ and f_a_ are attributed to the ΔE effects of magnetostrictive material as a function of the DC magnetic field. Specifically, the Young’s modulus *E*_m_ of magnetostrictive material varies with DC magnetic field due to the different magnetic domain movements under various magnetic fields. Since the resonance frequency of the ME composite is proportional to the square root of the ME composite’s average Young’s modulus $\bar{E}\;$($\bar{E}=n_{p}E_{p}+n_{m}E_{m},$ n_p_ and n_m_ are the volume fraction of piezoelectric and magnetostrictive materials, respectively) [22], the resonance frequency shifts with the increased DC magnetic field accordingly.

It is also shown in Figure 3 that the maximum impedance (Z_m_) and minimum impedance (Z_n_) vary nonmonotonically with the DC magnetic field. It is found that for Metglas/PZT composite, Z_m_ increases sharply until reaching a peak at H_dc_ = 40 Oe, then it gradually reaches a stable value with the further increased H_dc_. The variations of maximum and minimum impedances for both ME composites are due to both factors: on the one hand, when a DC magnetic field is applied along the length direction of the ME composite, the magnetostrictive material elongates and shrinks in the plane due to the piezomagnetic effect and ΔE effect. Then such magnetostrictive strain and stress transfer to the neighboring piezoelectric material due to the interface coupling, which lead to the varied dielectric polarization of ME composite with increasing H_dc_. On the other hand, the relative permeability of magnetostrictive material decreases with the increased DC magnetic field along the length direction since the large H_dc_ plays a damping role on the magnetic domain movement. In this case, the strong dependence of Z on the effective relative permeability $\mu_{\mathrm{eff}}$ and dielectric polarization $\varepsilon_{\mathrm{eff}}$ lead to the change of Z with the DC magnetic field. The detailed analysis is as follows.

According to the report by Salahun et al. [23], the impedance of ME laminate composite can be determined by Equation (1), as
(1)$$\;Z=\sqrt{\frac{\mu_{0}\mu_{\mathrm{eff}}}{\varepsilon_{0}\varepsilon_{\mathrm{eff}}}}$$
where $\mu_{0}$ and $\mu_{\mathrm{eff}}$ are vacuum permeability and relative effective permeability of magnetostrictive material, respectively. $\varepsilon_{0}$ and $\varepsilon_{\mathrm{eff}}$ are vacuum permittivity and relative effective permittivity of piezoelectric material, respectively. At low electrical excitation frequencies, the thickness of magnetostrictive material is far less than the skin depth, and the thickness of the piezoelectric material is far less than the sound wavelength in the material.

For ME laminate composite, $\mu_{\mathrm{eff}}$ and $\varepsilon_{\mathrm{eff}}$ can be calculated using Wiener’s law [24] as
(2)$$\;\;{\;\mu }_{\mathrm{eff}}=n_{m}\left(\mu_{r}-1\right)+1$$
(3)$$\;\;\varepsilon_{\mathrm{eff}}=\varepsilon_{r}/n_{p}$$
where $\mu_{r}$ and $\varepsilon_{r}$ are relative permeability of magnetostrictive material and relative permittivity of piezoelectric material, respectively.$\;n_{m}$ and $n_{p}$ are volume fractions of magnetostrictive and piezoelectric materials, respectively. Hence the relative permeability and permittivity of ME composite play key roles in tuning the impedance, according to Equations (1)–(3).

On the one hand, since the relative magnetic permeability of magnetostrictive material can be estimated as $\mu_{r}=4\pi M_{s}/H_{\mathrm{dc}}+1$, thus the magnetic permeability varies with the applied DC magnetic field H_dc_. From Equations (1) and (2), the strong dependence of effective relative permeability $\mu_{\mathrm{eff}}$ on H_dc_ results in the dependence of magnetoimpedance on H_dc_.

On the other hand, the relative permittivity varies with the change of applied mechanical stress T according to Devonshire’s law [25],
(4)$$\;\;\;\frac{1}{\;\varepsilon_{r}}=\frac{1}{\varepsilon_{r}\left(T=0\right)}-4Q_{12}T\;$$
where $Q_{12}$ is the electrostriction coefficient and T is the stress. When a DC magnetic field is applied, the magnetostriction of magnetostrictive material gives rise to the mechanical stress T, which transfers to the piezoelectric material due to the stress–strain coupling of the interlayer and further leads to varied dielectric polarizations of piezoelectric material. According to the report by Srinivasan et al. in [26], the corresponding mechanical stress T of the magnetostrictive layer can be expressed as following:(5)$$\;\;\;\;T=\frac{E_{p}E_{m}t_{m}\Delta s}{\left(1-\nu \right)\left(t_{p}E_{p}+t_{m}E_{m}\right)}$$ where ν=0.34 is Poisson’s ratio, Δs is the magnetostrictive strain of the magnetostrictive material. $E_{p}$ and $E_{m}$ are the Young’s modulus of piezoelectric and magnetostrictive material, respectively. $t_{p}$ and $t_{m}$ are the thickness of piezoelectric and magnetostrictive materials, respectively.

Substituting Equation (5) into Equation (4) yields
(6)$$\frac{1}{\varepsilon_{r}}=\frac{1}{\varepsilon_{r}\left(T=0\right)}-\frac{4Q_{12}E_{p}E_{m}t_{m}\Delta s}{\left(1-\nu \right)\left(t_{p}E_{p}+t_{m}E_{m}\right)}\;$$

From Equation (5), it is found that the mechanical stress T resulted from the magnetostrictive material varies with the DC magnetic field H_dc_ due to the dependence of the magnetostrictive strain Δs and Young’s modulus $E_{m}$ on H_dc_. Correspondingly, this leads to the change of $\varepsilon_{r}$ and impedance Z with the DC magnetic field according to Equations (1) and (6).

Then by inserting Equations (2) and (3) and Equation (6) into Equation (1), the ME composite’s impedance can be expressed as
(7)$$Z=\sqrt{\mu_{0}n_{p}\frac{n_{m}\left(\mu_{r}-1\right)+1}{\varepsilon_{0}}\;\left[\frac{1}{\varepsilon_{r}\left(T=0\right)}-\frac{4Q_{12}E_{p}E_{m}t_{m}\Delta s}{\left(1-\nu \right)\left(t_{p}E_{p}+t_{m}E_{m}\right)}\right]}\;\;\;$$

According to Equation (7), it is known that the variation of ME composite’s impedance (Z) with the magnetic field H_dc_ is not only determined by the relative magnetic permeability $\mu_{r}$ of the magnetostrictive material but is also affected by the magnetostrictive strain Δs and Young’s modulus $E_{m}$ of the magnetostrictive material since the magnetostrictive stress transferred to neighboring piezoelectric layer causes the varied $\varepsilon_{r}$. Hence, it is interesting to find that the magnetic permeability, magnetostrictive strain and Young’s modulus of the magnetostrictive layer play crucial roles in improving the magnetoimpedance effect of the ME composite. Here, it is noted that the mechanism of the MI effect in ME composites differs from the giant magnetoimpedance effect in soft magnetic materials [27]. The former is determined by both the permeability and permittivity of the ME composite, while the latter is mainly affected by the skin effect and magneto-inductance of soft magnetic material.

To further validate the theoretical analysis of magnetostrictive material properties’ impact on the magnetoimpedance effect, the normalized magnetization curves of Terfenol-D and Metglas were measured with the vibration sample magnetometer (VSM). Here, the DC magnetic field was applied along the longitudinal direction of the sample when the magnetization curves (Figure 4) were measured. The magnetization curve of Terfenol-D shows clear hysteretic behaviors compared with Metglas. The magnetization of Metglas reaches the saturation quickly with the increased H_dc_, while the magnetization of Terfenol-D varies more slowly with increasing H_dc_. This originates from the different structures of magnetic domains for Terfenol-D and Metglas, respectively. Nanosized striped domains in Metglas can be easily aligned along the direction of the applied DC magnetic field due to its small coercive field H_c_ and high reversibility. However, Terfenol-D possesses a larger coercive field relative to Metglas, which requires the larger DC magnetic field to reach the new magnetization state once the magnetic domains are reoriented and this leads to the hysteresis of the magnetization curve. Additionally, it is found that the maximum magnetic permeability, saturation magnetization and saturation magnetic field of Metglas are 50,000, 1016 emu cm^−3^ and 105 Oe, respectively. For comparison, the maximum magnetic permeability, saturation magnetization and saturation field of Terfenol-D are 10, 629 emu cm^−3^ and 3100 Oe, respectively. Here, the relative magnetic permeability $\mu_{r}$ of Metglas is 5000 times larger than that of Terfenol-D, which results in a significantly lower saturation field and produces higher magnetostrictive strain at low H_dc_ for Metglas.

It is well known that the Young’s modulus E of magnetostrictive material varies with applied DC magnetic field, i.e., ΔE effect ($\Delta E/E_{0}=\left(E_{H}-E_{0}\right)/E_{0}$, $E_{H}$ and $E_{0}$ are the elastic modulus in specific magnetic field H_dc_ and H_dc_ = 0 Oe, respectively [28,29].

Here, the variations of the elastic modulus E for the Metglas and Terfenol-D with a bias magnetic field are shown in Figure 5. For Terfenol-D, the elastic modulus E decreases slowly with increasing H_dc_ (negative-ΔE), which is mainly attributed to non-180° domain-wall motion. When H_dc_ increases to 373 Oe, the non-180° domain-wall motion achieves its maximum, and the compliance related to increased deformation is maximized, resulting in a minimum stiffness for Terfenol-D. Correspondingly, E reaches a minimum value. Then the elastic modulus E increases with further increasing H_dc_ (positive-ΔE) because the constraint of non-180° domain-wall motion at higher magnetic field tends to stiffen Terfenol-D. For comparison, the elastic modulus E of Metglas first decreases with the increased bias field and reaches a minimum value at a bias field of 3.5 Oe and the maximum negative ΔE effect occurs, then increases again until the magnetization of Metglas reaches the saturation and the positive ΔE effect happens. Hence, the dependence of Young’s modulus on H_dc_ results from the varied magnetic domain movement under changing H_dc_. It is noted that the maximum absolute value ${\left(\Delta E/E_{0}\right)}_{max}$ of Metglas is ~17.5% at low magnetic field H_dc_, which is about two times larger than that of Terfenol-D. This is mainly attributed to the high saturation magnetic field of Terfenol-D. It is known that the ΔE effect is related to saturation magnetostriction and the saturation magnetization of the magnetic material, which can be expressed as [30]
(8)$$\frac{\Delta E}{E_{0}}=\frac{9\mu_{0}E_{H}\lambda_{s}^{2}}{20{\pi M}_{s}^{2}}$$
where $\mu_{0}$ is the vacuum permeability, $\lambda_{s}$ is the saturation magnetostriction, and $M_{s}$ is the saturation magnetization. $\Delta E/E_{0}$ is not only determined by $\lambda_{s}$ but is also affected by $M_{s}$. Hence, Terfenol-D with giant saturation magnetostriction will exhibit the large ΔE effect. However, such a large ΔE effect can be achieved only at the extremely high magnetic field due to the high saturation magnetization and magnetocrystalline anisotropy of Terfenol-D. Clark et al. reported that the largest $\Delta E/E_{0}$ effect for Terfenol-D reaches 161% at the high DC bias magnetic field of 4.3 kOe [30].

Additionally, the magnetostrictive strain coefficients d_33_ of Terfenol-D and Metglas at various H_dc_ are measured with a Laser Doppler Vibrometer LDV system, respectively, as shown in Figure 6. For the Terfenol-D, d_33_ enhances slowly and reaches a maximum value when H_dc_ increases to 373 Oe and then reduces subsequently with the further increased H_dc_. For the Metglas, d_33_ increases linearly with increasing H_dc_ until reaching a maximum value at H_dc_ = 3.5 Oe, and then drops rapidly to a near-zero value when H_dc_ further increases. It is interesting to find that the maximum d_33_ (Figure 6) occurs at the same H_dc_, where the maximum negative ΔE effect (Figure 5) also happens. It indicates that the magnetostrictive materials are driven in the “burst region” of the quasi-static strain-field curve where non-180° domain-wall motion is maximum. Furthermore, it is found that the effective magnetostrictive strain coefficient d_33_ of Metglas is greater than that of Terfenol-D in a small DC magnetic bias range of 0 < H_dc_ < 33 Oe. Specifically, the maximum value d_33_ of Metglas is about 1.2 times larger than that of Terfenol-D due to the small saturation field. This is because the magnetostrictive strain coefficient d_33_ is directly proportional to the saturation magnetostriction λ_s_ and the squared magnetic relative permeability $\mu_{r}$ [31,32,33]. Although the saturation magnetostriction of Terfenol-D (λ_s_ = 1200 ppm) is larger than that of Metglas (λ_s_ = 20 ppm), Terfenol-D presents a quite low relative magnetic permeability (μ_r_ = 10), and correspondingly, the high permeability of Metglas causes its large magnetostrictive strain coefficient at the lower DC magnetic biases.

It is known that the impedance of ME composite is determined by the effective magnetic permeability, Young’s modulus and magnetostrictive strain of magnetostrictive material, shown as Equation (7). Correspondingly, much higher permeability, larger magnetostrictive strain, stronger ΔE effect and smaller saturation magnetic field of Metglas result in larger magnetoimpedance ratios (i.e., ΔZ/Z) for Metglas/PZT composite compared to Terfenol-D/PZT.

The magnetoimpedance ratio (MR) is defined as Equation (9) to characterize the MR effect [27,34]
(9)$$\frac{\Delta Z}{Z}\;=\;\frac{Z\left(H_{\mathrm{dc}}\right)-Z_{\min}}{Z_{\min}}$$
where $Z\left(H_{\mathrm{dc}}\right)$ denotes the impedance of ME composite at H_dc_ and $Z_{\min}$ denotes the minimum impedance.

For comparison, we measure impedance spectra, the minimum impedance Z_n_ and maximum impedance Z_m_ as a function of the DC magnetic field H_dc_ for Terfenol-D/PZT composite, as shown in Figure 7. For Terfenol-D/PZT composite, Z_m_ decreases rapidly with the increased H_dc_ until reaching the minimum value near H_dc_ = 300 Oe, then Z_m_ increases again with further increased *H*_dc_. Then, the MRs at the resonance f_r_ and antiresonance f_a_ are measured as a function of H_dc_ for Terfenol-D/PZT and Metglas/PZT composites, respectively, shown as Figure 8. It is found that the maximum ΔZ/Z of Terfenol-D/PZT composite is about 69.84% and 43.98% at the antiresonance frequency f_a_ and resonance frequency f_r_, respectively. For comparison, the maximum magnetoimpedance ratios ΔZ/Z of Metglas/PZT composite are about 605.24% and 239.98% at the antiresonance and resonance frequencies, respectively. Compared with the Terfenol-D/PZT composite, the MRs of Metglas/PZT composite are much higher at both the resonance and antiresonance frequencies, respectively. For example, the maximum ΔZ/Z of Metglas/PZT composite is 8.6 times as high as that of Terfenol-D/PZT at the antiresonance frequency. The reason is as follows: although Metglas exhibits much smaller saturation magnetostriction compared to Terfenol-D, the extremely high permeability of Metglas (μ_r_ = 50,000) concentrates the external magnetic flux effectively and results in the high effective strain coefficient and ΔE effect in a low magnetic field, shown as Figure 5 and Figure 6. Such strong magnetostrictive strain and ΔE effect at the low H_dc_ cause the sharply varied effective dielectric permittivity of the neighboring piezoelectric layer through the stress–strain coupling. Additionally, the relative magnetic permeability of Metglas decreases more sharply with the increased DC magnetic field due to the low saturation field. Correspondingly, a larger magnetoimpedance ratio is obtained for the Metglas/PZT composite due to the drastic change of permeability and permittivity. In contrast, even though Terfenol-D possesses a huge saturation magnetostriction (*λ**_s_* = 1200 ppm) and low Young’s modulus, the low relative permeability of Terfenol-D means that a significantly higher magnetic bias field is needed to generate the large magnetostrictive strain and ΔE effect, resulting in the small variation of effective dielectric permittivity at the low bias field. Furthermore, the magnetic permeability of Terfenol-D varies more slowly with the bias magnetic field compared to Metglas. Hence, this leads to a smaller magnetoimpedance ratio of Terfenol-D/PZT composite relative to Metglas/PZT composite.

## 4. Conclusions

In this study, the magnetoimpedance effects are investigated for bilayer ME composites with different magnetostrictive materials (i.e., soft magnetic amorphous ribbon Metglas and giant magnetostrictive material Terfenol-D). Both the theoretical analysis and experimental studies show that the magnetic permeability, magnetostrictive strain and Young’s modulus of the magnetostrictive layer play a crucial role in the magnetoimpedance effect of ME composite. Although Metglas possesses a lower saturation magnetostriction relative to Terfenol-D, the magnetic permeability of Metglas is significantly larger than that of Terfenol-D. This leads to the larger variation of effective magnetostrictive strain at the low bias field and corresponding larger MI effect. The experimental results show that the maximum magnetoimpedance ratio of Metglas/PZT composite is about 605.24% at the antiresonance frequency, which is 8.6 times as high as that of Terfenol-D/PZT. The study indicates that the magnetoimpedance effect can be improved significantly by utilizing the magnetostrictive material with optimum material properties, which provides guidance for designing a magnetically tunable electrical device.

## Figures and Tables

**Figure 1 materials-14-06397-f001:**
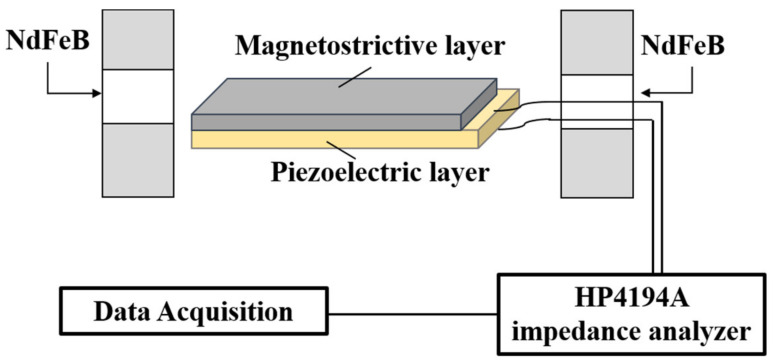
Schematic diagram of the dynamic measurement setup for investigating the MI effect of ME composite.

**Figure 2 materials-14-06397-f002:**
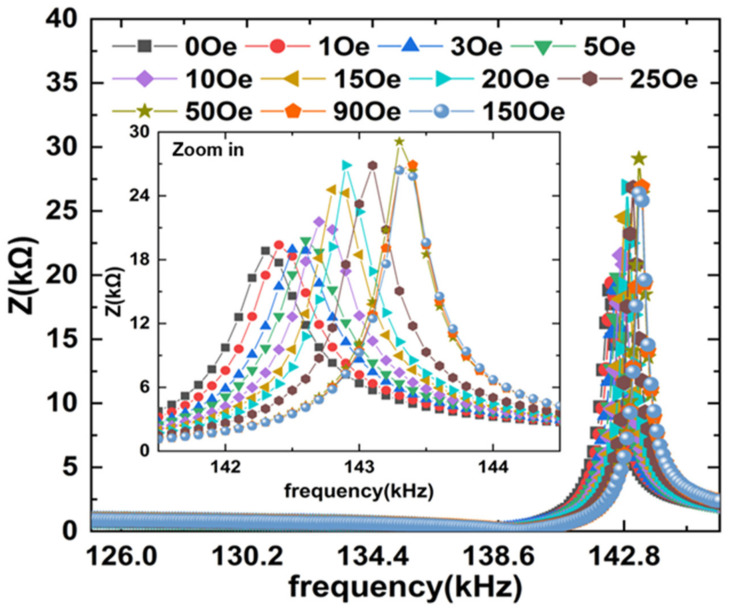
Impedance spectra of the bilayer Metglas/PZT composites at various DC magnetic fields. The inset shows the enlarged view near the antiresonance frequency.

**Figure 3 materials-14-06397-f003:**
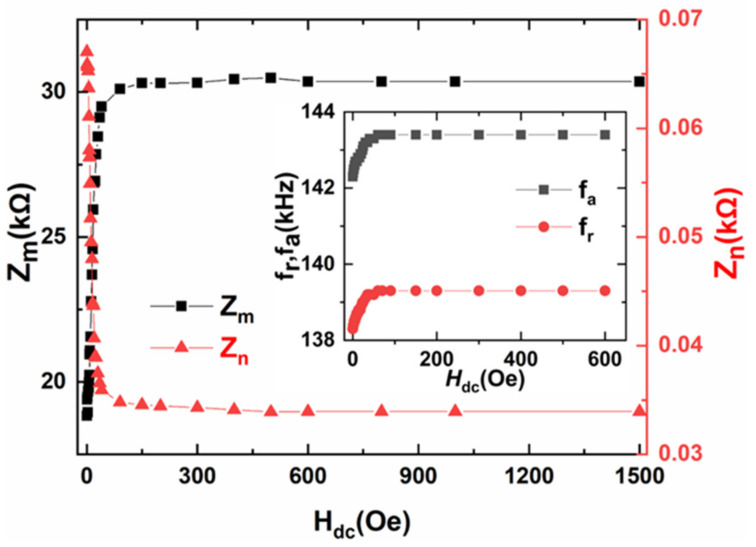
The minimum impedance Z_n_ (@the resonance frequency f_r_) and maximum impedance Z_m_ (@the antiresonance frequency f_a_) as a function of the DC magnetic field H_dc_ for bilayer Metglas/PZT composites. The inset shows the dependence of f_r_ and f_a_ on H_dc_.

**Figure 4 materials-14-06397-f004:**
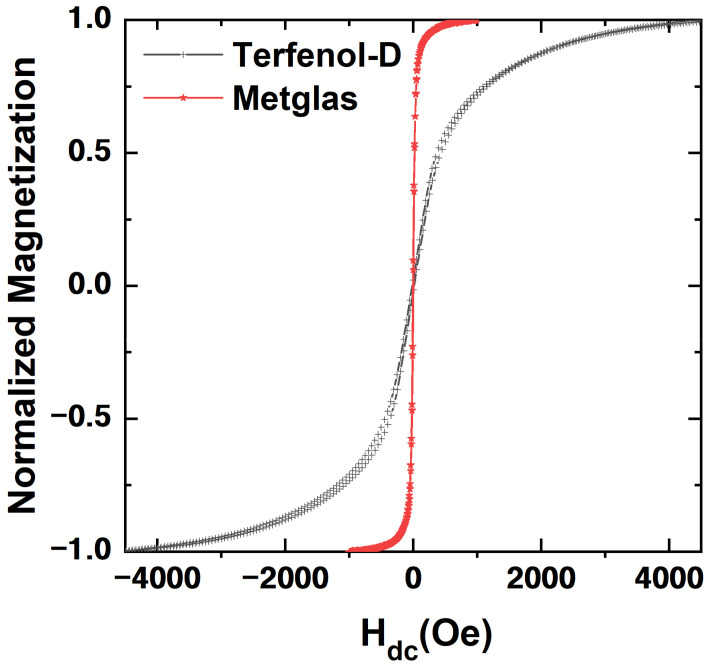
The measured magnetic hysteresis loops of the Metglas and Terfenol-D, respectively.

**Figure 5 materials-14-06397-f005:**
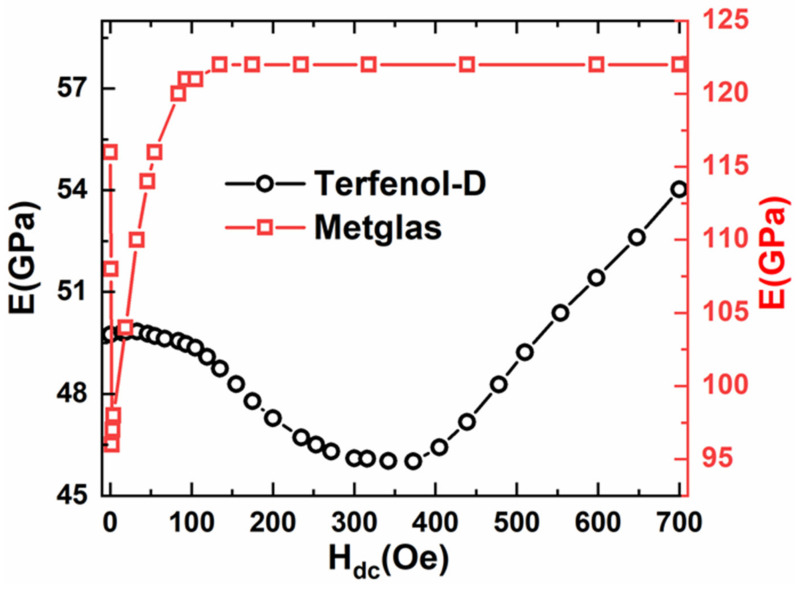
The elastic modulus E as a function of H_dc_ for the Metglas and Terfenol-D.

**Figure 6 materials-14-06397-f006:**
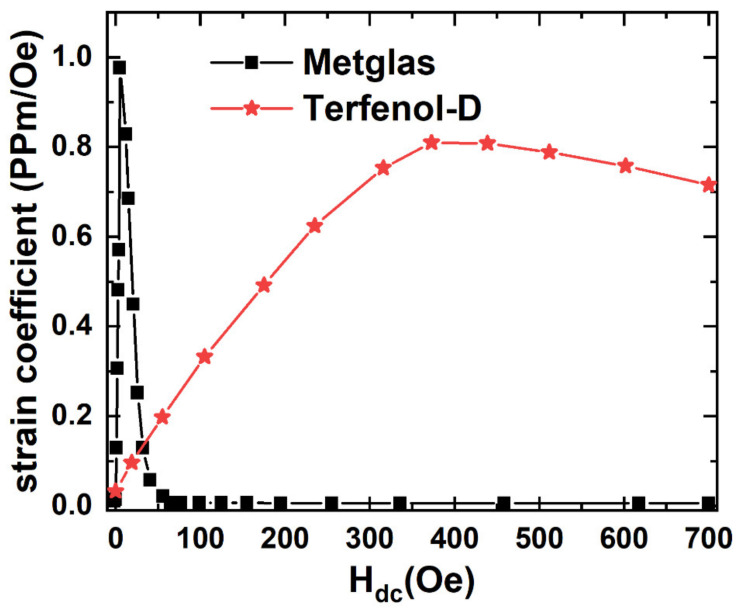
The strain coefficients as a function of H_dc_ for the Metglas and Terfenol-D.

**Figure 7 materials-14-06397-f007:**
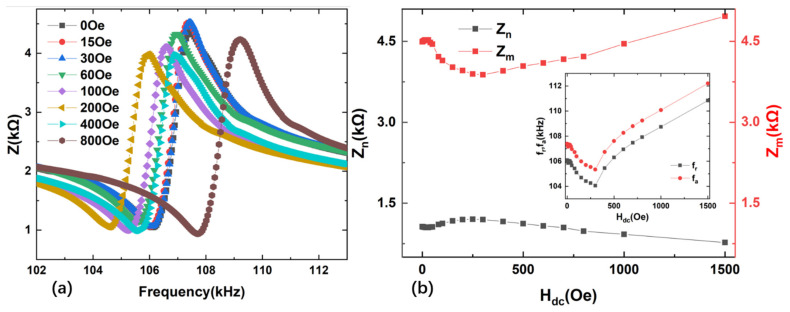
(**a**) Impedance spectra of the Terfenol-D/PZT composites at various DC magnetic fields. (**b**) The minimum impedance Z_n_ and maximum impedance Z_m_ as a function of the DC magnetic field H_dc_ for Terfenol-D/PZT composites. The inset shows the dependence of f_r_ and f_a_ on H_dc_.

**Figure 8 materials-14-06397-f008:**
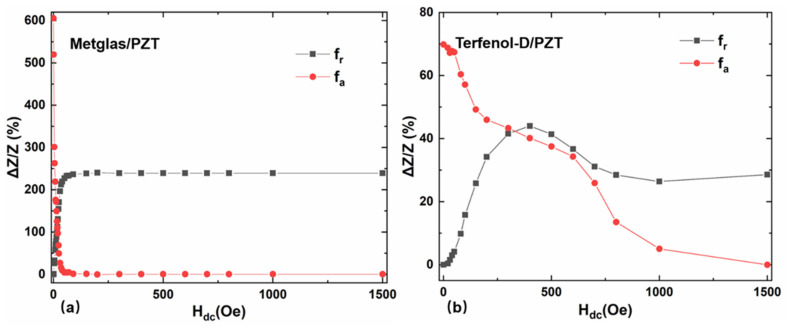
The magnetoimpedance ratios as a function of the DC magnetic field for (**a**) Metglas/PZT and (**b**) Terfenol-D/PZT composites at f_r_ and f_a_, respectively.

## Data Availability

Data sharing is not applicable to this article.

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
