# Peer review of "The Study of Magnetoimpedance Effect for Magnetoelectric Laminate Composites with Different Magnetostrictive Layers"

_materials, 2021, doi:10.3390/ma14216397_

Round 1

Reviewer 1 Report

The paper "The study of magnetoimpedance effect for magnetoelectric laminated composites with different magnetostrictive layers" is well written and brings interesting and useful experimental investigations. 

I have some remarks:

  • page 3, line 92: the magnetic field is given in mT (up to 1500), but in further text it is in Oe (up to 1500 as well)... this is probably not  correct.
  • page 6, lines 206, 208: "relative magnetic permeability" - do you mean the maximum one? (because magnetic permeability varies with dc magnetic field) 
  • page 7, line 219:  should be "Figure 5" instead of "6"
  • Figure 6: Y-axis caption - should be "strain" instead of "stain"

Author Response

Dear Editor, 

 Thank you very much for your letter dated October 15, 2021, and the reports from the reviewer about our paper submitted to Materials. (Article ID: materials-1419035). The whole reports have been fair, encouraging and constructive. We have learned much from them.

 After carefully studying the reviewer’s reports, we have made corresponding revisions to the manuscript according to the reports of reviewer. We have also responded point by point to reviewer’s reports as listed below, along with clear indications of the location of the revision. 

If you have any question about this paper, please don’t hesitate to contact me. 

We hope these will make it more acceptable for publication. 

Sincerely yours, 

The corresponding author:

Yao Wang

Phd, Associate Professor

School of Electronic Information and Electric Engineering,

Shanghai Jiao Tong University,

Shanghai 200240, China

Tel: +86-13677620870

Reviewer 2 Report

Overall, this work is scientifically sound, with clear conclusions supported by convincing experimental data and theoretical explanations.

However several issues should be resolved prior to publication:

  • Many spelling/grammatical errors appear in the text, raising doubts about careful proofreading before submission
  • Style and English should globally be improved
  • The terms "obvious" or "obviously" should be used sparingly
  • Figure 6 is referenced in the text prior to figure 5
  • The formulation in lines 175 to 178 is awkward: Eq. 6 does not show that T varies with Hdc ("From eq. 6 it is found...") but rather (as indeed explained in the following sentence) that εr varies with Hdc as a result of the variation of T.
  • The authors claim that "few articles have discussed about tuning the giant magnetoimpedance (MI) effect of ME composites with the dc magnetic field at room temperature." (line 48), but omit citing at least one of these articles (for instance Leung et al., "Enhanced tunability of magneto-impedance and magneto-capacitance in annealed Metglas/PZT magnetoelectric composites", AIP Advances 8, 055803 (2018))
  • As the authors focus on the comparison between Metglas and Terfenol-D, the impedance spectra, Zm, Zn, fr, and fa , and  of the Terfenol-D/PZT composite should ideally be presented as well
  • Even though experimental data support the conclusions, a comparison between theoretical and experimental magnetoimpedance ratios would be welcome.

As a result, I recommend accepting this article after minor revisions.

Author Response

Dear Editor, 

Thank you very much for your letter dated October 15, 2021, and the reports from the reviewer about our paper submitted to Materials. (Article ID: materials-1419035). The whole reports have been fair, encouraging and constructive. We have learned much from them.

 After carefully studying the reviewer’s reports, we have made corresponding modifications to the manuscript. The relevant regulations had been made in the original manuscript according to the reports of reviewer. We also responded point by point to reviewer’s reports as listed below, along with a clear indication of the location of the revision. 

If you have any question about this paper, please don’t hesitate to contact me. 

We hope these will make it more acceptable for publication. 

Sincerely yours, 

The corresponding author:

Yao Wang

Phd, Associate Professor

School of Electronic Information and Electric Engineering,

Shanghai Jiao Tong University,

Shanghai 200240, China

Tel: +86-13677620870

Reviewer 3 Report

The paper is wery well organized and presented.

The aim of the work is finely defined in the introduction, together with a sufficiently detailed overview on the state of the art.

In my opinion the choice of the experimental activities was well conducted in order to support the work and it's conclusions; a solid theory contribution helps to complete the logical path.

Only very few minor typos (e.g. "laminate" in place of "laminated") are present in the text, which could be revised by the authors.

The English language and style are fine.

I appreciated the use of zooming windows in graphs (e.g. in figure 2). An option could be to use it in figure 3 also, but since a child window with the interesting resonance and antiresonance frequency shift is still present, I understand your chioce of presentation.

Author Response

(The authors gave the same response as above.)
